# Parietal-specific activation reveals pain from inadequate levels of altitude acclimatization/ adaptation

**Shurong Jia**[1,2], **Niannian Wang**[1,2], **Rui Su**[1,2,3], **Hailin Ma**[1,2]*, **Hao Li**[1,2]*

**1** Tibet Autonomous Region Key Laboratory for High Altitude Brain Science and Environmental Acclimatization, Tibet University, Lhasa, China, **2** Plateau Brain Science Research Center, Tibet University, Lhasa, China, **3** School of Psychological and Cognitive Sciences and Beijing Key Laboratory of Behavior and Mental Health, Peking University, Beijing, China

\* futanghu888@126.com (HL); 83976475@qq.com (HM)

## Abstract

### Objective

Pain is acknowledged as a prominent physiological symptom of high-altitude (HA) acclimatization, yet there exists a dearth of empirical support regarding the impact of variations in HA acclimatization/ adaptation on pain perception in hypoxic environments.

### Methods

A total of 65 HA residents were recruited, all of whom had resided continuously in HA regions for a minimum duration of one month. The study involved an assessment of peripheral oxygen saturation ($SpO_2$) and haematocrit (HCT) levels, administration of the Pain Frequency, Intensity and Burden Scale (P-FIBS), and the resting-state electroencephalogram (rs-EEG) signals of all frequency bands including delta, theta, alpha, beta and gamma were acquired from the frontal lobe, parietal lobe, temporal lobe and occipital lobe. Residents' acclimatization levels were assessed using altitude acclimatization/ adaptation index (AAI), and categorized into high and low AAI groups.

### Results

Residents in the high-AAI group reported lower levels of perceived pain compared to those in the low-AAI group ($t = 1.61$, $p = 0.04$). Further analysis using EEG frequency bands demonstrated that parietal delta ($\beta = -0.35$, 95%CI = [−1.18, −0.01]) and beta ($\beta = 0.31$, 95%CI =[0.01, 1.19]) powers acted as mediators in the relationship between AAI and pain perception. Brain waves of other frequencies, including theta ($\beta = -0.02$, 95%CI = [−0.46, 0.41]), alpha ($\beta = -0.03$, 95%CI = [−0.34, 0.18]), and gamma ($\beta = -0.06$, 95%CI =[−0.09, 0.03]), did not show a significant mediating effect.

**Data availability statement:** The data contain sensitive patient information and may not be shared publicly. Data access is restricted by the Ethics Committee of Tibet University. Researchers eligible for access to confidential data can contact Hailin Ma (83976475@qq.com), a member of the Ethics Committee of Tibet University, or Hao Li (futanghu888@126.com), corresponding author of this article.

**Funding:** This research was supported by Key research and development projects in Tibet Autonomous Region (grant number XZ202201ZY0048G); National Natural Science Foundation Regional innovation development joint fund project of China (grant number U23A20476); the National Natural Science Foundation of China (grant number 32260212); the project of the Tibet Autonomous Region Science and Technology Program "Unveiling the List of Commander-in-Chief", Key research and development projects in Tibet Autonomous Region (grant number 2023ZYJM001).

**Competing interests:** No authors have competing interests

## Conclusions

A lower level of HA adaptation indicates a higher intensity of pain perception. The enhancement of EEG activity might be regarded as a compensatory mechanism. It endeavors to maintain the normal operation of the body's physiological functions by increasing the efficiency of neural activities, thereby helping individuals to improve their adaptability to the HA environment. The parietal lobe, as a key brain region responsible for processing sensory information throughout the body, exhibits the most significant activation state compared with other brain regions when an individual is in a HA environment and the adaptation level shows dynamic changes. Notably, once the beta waves, which are highly associated with attention, are activated, the increase in an individual's alertness often indicates that the individual will experience a more intense and higher level of pain sensation. However, the activation of delta waves, which usually occur during the stage of deep sleep and can prompt the brain to maintain a relaxed state, generally means that the individual will perceive a relatively lower level of pain. Consequently, the rs-EEG power in the delta and beta frequency bands of the parietal lobe region serves as an important mediating factor in the connection between AAI and pain perception.

## 1. Introduction

High altitude (HA) presents a variety of environmental challenges, including hypoxia, cold, intense solar radiation, high wind speeds, and low air humidity. Among these factors, low-pressure hypoxia is particularly noteworthy as it poses a significant obstacle for human adaptation to HA environments. This challenge leads to various physiological responses in HA residents, such as increased heart rate, elevated blood pressure, myocardial contraction, and respiratory difficulties [1,2]. Furthermore, living in HA conditions also results in notable alterations in sensory functions [3–7]. Somatic nociception is a common physiological response experienced upon exposure to high altitudes, characterized by pain resulting from stimulation of the trunk, including diffuse muscle pain [8], abdominal pain [9], neck pain [10], orofacial pain [11,12], headache [13,14], earache [15], and arthralgia [16]. The brain is susceptible to disruptions in aerobic metabolism in the hypoxic conditions of HA, resulting in diminished cognitive performance, heightened negative affect [17,18], and increased pain perception stemming from aberrations in sensory processing [19–21]. This pain experience is linked to central sensitization, a process characterized by heightened excitability of central neurons in response to repetitive stimuli, rendering them hypersensitive to both noxious (e.g., hypoxic stress) and non-noxious stimuli [22]. Exposure to HA hypoxia triggers a transient or sustained elevation in the steady-state levels of reactive oxygen species (ROS), thereby disrupting cellular metabolism and signaling pathways, elevating the stress level [23], intensifying the neuro-inflammatory response [24,25], ultimately leading to the sensation of pain.

Physiological adjustments in response to prolonged hypoxic conditions, referred to as "altitude acclimatization/ adaptation" [26], include a slight elevation in erythrocyte count and hemoglobin levels [27], enhance antioxidant defenses, and preserve redox balance [28,29]. Studies have demonstrated that in HA residents, there are significant temporal differences in the changes of oxidative stress markers and antioxidant enzyme levels [30–32]. Pain perception has been linked to inflammatory response [33] and CNS-neuroimmunity [34]. In the absence of sustained stimulation, the CNS's heightened sensitivity to pain perception returns to its baseline level [35], highlighting the impact of altitude acclimatization/ adaptation on sensory function, suggesting the existence of a response pathway involving altitude acclimatization/ adaptation levels, hypoxic stress, neuroinflammatory response, and central sensitization leading to altitude-induced pain.

Furthermore, the physical discomfort resulting from low altitude acclimatization/ adaptation is characterized by nociceptive sensations induced by a hypoxic environment, not pathological pain [36]. The hypoxic environment-induced inflammatory response exacerbates the release of pro-inflammatory factors, leading to the disruption of the blood-brain barrier and subsequent infiltration of these factors into the brain, ultimately resulting in a disorder of the brain microenvironment and activation of neuroimmunity [37]. This process is associated with decreased synaptic signaling function and altered brain structural function [38]. Consequently, it is hypothesized that individuals with inadequate acclimatization/ adaptation to sustained altitude hypoxia may experience heightened oxidative stress and neuroinflammatory responses, resulting in increased central sensitivity and perception of pain.

Sensory information, including pain signals, is transmitted from receptors to the parietal lobe through the spinal tract and thalamus [39], the parietal cortex, in conjunction with the thalamus, collaborates in the processing of this information [40]. The parietal cortex serves as a crucial brain region for the integration of diverse sensory inputs, particularly in the realm of somatosensory perception [39–43]. Moreover, the perception of pain correlates with varying frequencies of neuronal oscillation and synchronization. Research employing animal pain models has demonstrated that an increase in theta oscillations leads to the alleviation of pain sensation [44–46]. Alpha waves' oscillatory intensity serving as a dependable measure for discerning pain perception and subjective pain levels [47–52]. Studies of gamma, beta, and delta waves have also revealed a correlation with pain [49,53–55], found that beta activity may serve as an indicator of changes in pain intensity [54,56]. Additionally, Hu and Iannetti [57] have proposed that gamma could potentially reflect variations in individual pain sensitivity and serve as a distinguishing feature of neuropathic pain, as suggested by Zhou et al [58]. Gamma oscillations have been found to be connected to the inhibitory control system and are associated with various aspects of pain, such as pain perception [59], pain intensity [60], and attention to pain [61]. Therefore, we proposed the following two hypotheses. H1: lower levels of altitude acclimatization/ adaptation are associated with heightened pain sensations. H2: brain oscillations in the five bands of the parietal lobe (delta, theta, alpha, beta, and gamma) may play a mediating role in the relationship between altitude acclimatization/ adaptation and altitude-related pain.

It endeavors to identify the correlation between altitude acclimatization/ adaptation and pain, disclose the physiological impacts of acclimatization/ adaptation on pain perception, and establish a foundation for novel non-pathological pain management strategies in individuals residing at HA.

## 2. Methods

### 2.1. Participants

We recruited 70 right-handed participants from September 15, 2023 to October 15, 2024, comprising Tibetans who have lived in HA areas for generations and Han people who migrated to HA regions. They had been studying or living in Lhasa (at an altitude of 3680 m) continuously for over one month. Five participants were excluded from the electroencephalogram (EEG) analysis. Among them, three were excluded because excessive movements and muscle artifacts contaminated more than 50% of the trials, and two were excluded due to incomplete physiological data. Eventually, 65 participants (31 F, 34 M, age = 22.22 ± 2.43) were included in the analysis (Table 1). All the participants were healthy, free from neurological or psychiatric disorders, brain injuries and drug addictions. Prior to the experiment, none of the participants had taken painkillers

**Table 1. The basic characteristic information of participants (*N* = 65).**

|  | Age | Duration | AAI | Delta | Theta | Alpha | Beta | Gamma | Total | P-FIBS |
|---|---|---|---|---|---|---|---|---|---|---|
| *M ± SD* | 22.22 ± 2.43 | 3.43 ± 2.23 | 1.94 ± 0.37 | 3.43 ± 2.36 | −1.58 ± 2.60 | 1.63 ± 4.22 | −6.68 ± 2.82 | −13.77 ± 3.35 | −16.97 ± 13.35 | 1.29 ± 0.49 |
| Median | 22.00 | 3.00 | 1.91 | 3.45 | −1.81 | 1.81 | −6.99 | −14.56 | −18.85 | 1.00 |
| Lowest | 18.00 | 0.00 | 2.00 | −2.00 | −6.00 | −7.00 | −13.00 | −20.00 | −40.00 | 1.00 |
| Highest | 31.00 | 12.00 | 4.00 | 9.00 | 5.00 | 10.00 | 0.00 | −3.00 | 17.00 | 3.00 |

Note: M-mean, SD-standard deviation.

or other medications that might interfere with pain perception or facilitate altitude acclimatization/ adaptation. All participants gave their written informed consent and were compensated for their participation. The study was approved by the Ethics Committee of Tibetan University and followed the Declaration of Helsinki.

## 2.2. Recording equipment

EEG data were acquired utilizing a NeuroScan Curry 7 system, with electrodes positioned in accordance with the international 10–20 standard. The sampling frequency was set at 500 Hz, and all electrodes were verified to have an impedance of less than 5 kΩ. Signal amplifiers were utilized to enhance the signals and enable uninterrupted EEG recording. Online filtering was conducted within a bandwidth of 0.01 to 100 Hz during a 4-minute resting state with eyes closed.

## 2.3. Data pre-processing

EEGLAB, an open-source toolbox within the MATLAB environment (version 2021b, MathWorks, Inc., Natick, MA), was utilized for preprocessing EEG data [62]. The continuous EEG data underwent high-pass filtering at 1 Hz, while a low-pass filter with a cutoff frequency of 80 Hz was applied using a primary FIR filter. Electrodes deemed potentially undesirable were identified and corrected through spherical interpolation. Subsequently, data were segmented into two time periods, with the contaminated time period manually labeled and excluded from further analyses.

rs-EEG data contaminated by various sources of physiological, including muscle contraction and cardiac signal and environmental noise were processed using independent component analysis (ICA) with the runica algorithm as described by Jung et al [63]. Components with a probability greater than 0.7, identified using the ICLabel plug-in developed by Pion-Tonachini, Kreutz-Delgado and Makeig [64], were removed from the data, resulting in an average removal of 5 components per subject with a standard deviation of 2.0. Epochs with amplitudes exceeding ±100 μv were also excluded. Subsequently, the rs-EEG data were re-referenced to the average reference to complete the preprocessing of the data series.

## 2.4. Analysis of EEG

Power Spectral Density (PSD) was analyzed for each participant and EEG epoch based on the brain region segmentation outlined by Chen et al [65]. These measurements encompassed the entire brain, including all channels, were focused on six regions of interest (ROI): (i) frontal lobe (FP2, Fz, FP1, F3, F7, F8, F4), (ii) frontal-central region (Cz, FC1, C3, C4, FC2), (iii) parietal lobe (CP1, P3, Pz, CP2, P4), (iv) left temporal-parietal junction (FC5, FT9, T7, CP5, P7), (v) right temporoparietal junction (P8, CP6, T8, FT10, FC6), and (vi) occipital lobe (O1, Oz, O2). The analysis involved examining the power of each channel within specific frequency bands, including delta (δ; 1–3 Hz), theta (θ; 4–7 Hz), alpha (α; 8–13 Hz), beta (β; 14–30 Hz) and gamma (γ; 31–80 Hz). A spectral resolution of 250 points per channel was achieved by applying a Welch function to quantify the PSD using a 4 s Hamming window with a 50% overlap between successive windows. The data were then log-transformed using a $10*\log^{10}$ scale. Subsequently, brainwave metrics were computed for each channel, averaged across the entire brain and each ROI.

## 2.5. Pain sensation measurement

Pain sensation was assessed using the P-FIBS (pain frequency, intensity, and burden scale [66]. This scale comprises four items that measure the frequency, intensity, and burden of pain. Each item is rated on a 0–8 Likert scale, with lower scores indicating lower levels of pain or burden experienced in the previous week. This self-rating scale demonstrates high internal consistency (Cronbach's alpha coefficient = 0.90) and robust construct validity, making it a reliable instrument for assessing pain perception in individuals.

## 2.6. Altitude acclimatization/ adaptation measurement

Altitude acclimatization/ adaptation index (AAI, $SpO_2$/ HCT) was utilized to assess the degree of HA acclimatization/ adaptation in individuals [26]. A higher value of this index, reflecting higher $SpO_2$ and/ or lower HCT, indicates better acclimatization/ adaptation to the HA environment. Blood samples were obtained via venous blood sampling and analyzed for HCT at a hospital located in Lhasa (3680 m), China. $SpO_2$ was measured by a finger-clip oximeter (YUWELL YX306, Jiangsu Yuyue Medical Equipment & Supply Co., Ltd., Jiangsu).

## 2.7. Statistical analyses

Data analysis was performed using SPSS version 21.0. Given the central aim of the study—to investigate the intrinsic relationship between HA adaptation and pain—the AAI index was chosen as the main physiological marker to reflect the level of HA acclimatization/adaptation. Pain was quantified using the P-FIBS. An independent-samples t-test was conducted to compare pain levels across different levels of HA adaptation. Pearson correlation analysis was used to assess the strength of association between the AAI index and P-FIBS scores, with significance determined at $p < 0.05$. Drawing on prior theoretical frameworks and the present study's rationale, EEG activity in various parietal lobe frequency bands—due to its potential key role in the neural pathway between HA adaptation and pain perception—was introduced as a mediating variable. A regression-based mediation analysis was conducted using the following models:

Model 1: AAI as the independent variable and pain as the dependent variable (total effect);

Model 2: AAI as the independent variable and EEG activity as the dependent variable (effect on the mediator);

Model 3: both AAI and EEG activity as predictors, with pain as the dependent variable, to estimate the direct and indirect effects. To improve the robustness of the mediation estimates, the Bootstrap method was applied with 5000 resamples. This approach yielded an empirical distribution of the mediation effect and its 95% confidence interval. A confidence interval that does not include zero indicates a statistically significant mediation effect.

## 3. Results

### 3.1. Comparison of differences in pain sensation at the AAI

Based on the AAI scores, the participants were divided into two categories, with 32 residents in the high-AAI group and 33 residents in the low-AAI group, and the pain sensation was compared to the independent samples t-test for difference, which showed that there was a significant difference (t = −1.04, $p = 0.04$) in pain sensation between the low-AAI group (2.24 ± 1.54) and the high-AAI group (2.94 ± 1.93), indicating that the low-AAI group perceived pain more intensely than the high-AAI group (Table 2).

### 3.2. Correlation between AAI, parietal EEG and pain

The study examined the relationship between individuals' AAI, parietal rs-EEG at various frequencies, and pain (Figs 1 and 2). The results demonstrated a series of significant correlations. Firstly, a significant negative correlation was found between AAI and pain (r = −0.26, $p = 0.04$). Second, a significant negative correlation was observed between pain and δ frequency activity (r = − 0.31, $p = 0.01$), suggesting that greater δ activity was linked to reduced pain. Finally, AAI was

 

**Table 2. Differences in pain perception under different levels of AAI.**

| | Low-AAI (*N* = 33) | High-AAI (*N* = 32) | *t* |
|---|---|---|---|
| Pain | 2.24 ± 1.54 | 2.94 ± 1.93 | 1.61* |

Note: $^*p < 0.05.^{**}p < 0.01.^{***}p < 0.001.$

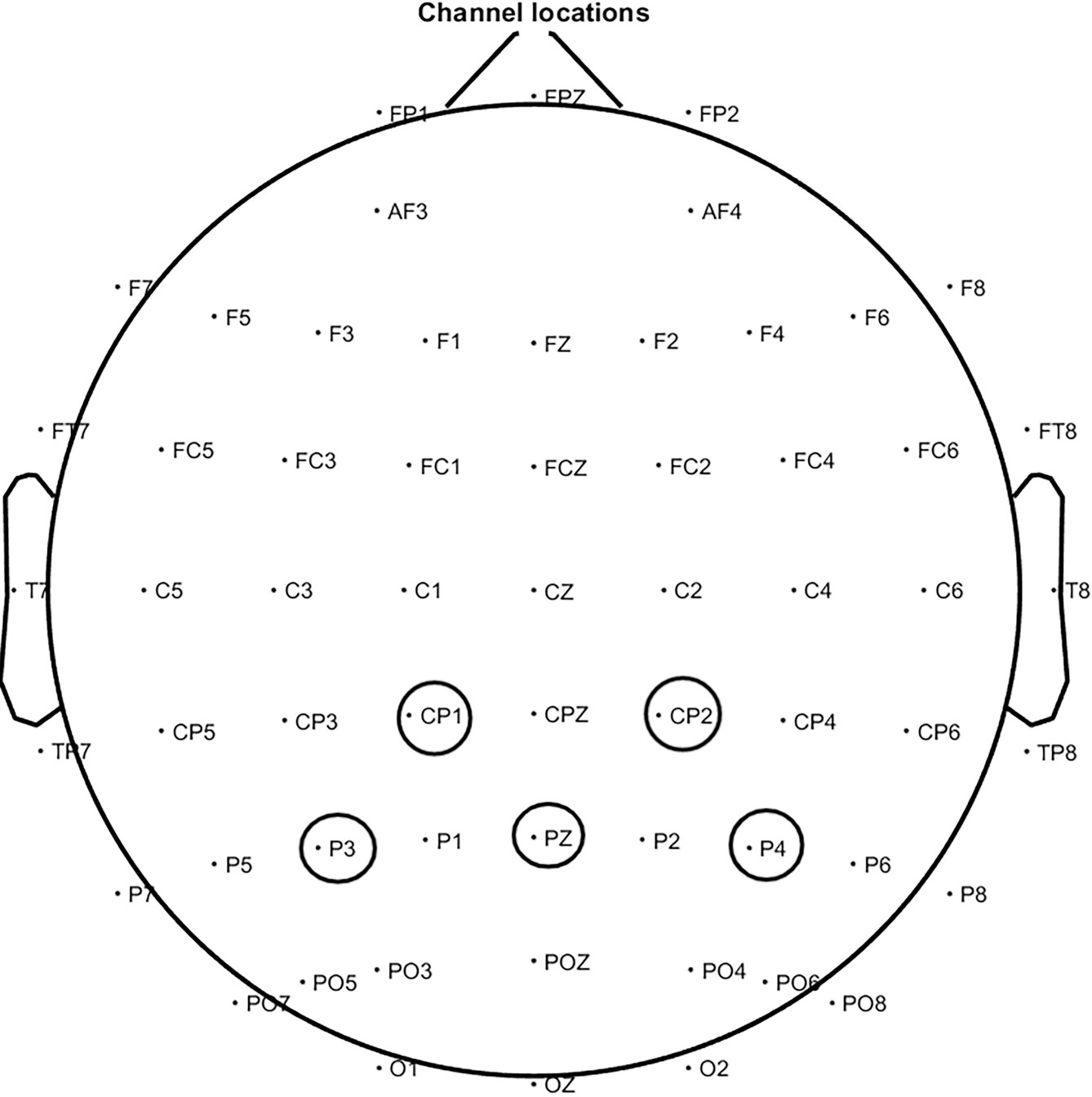

**Fig 1. Electrical power diagram of parietal lobe.**

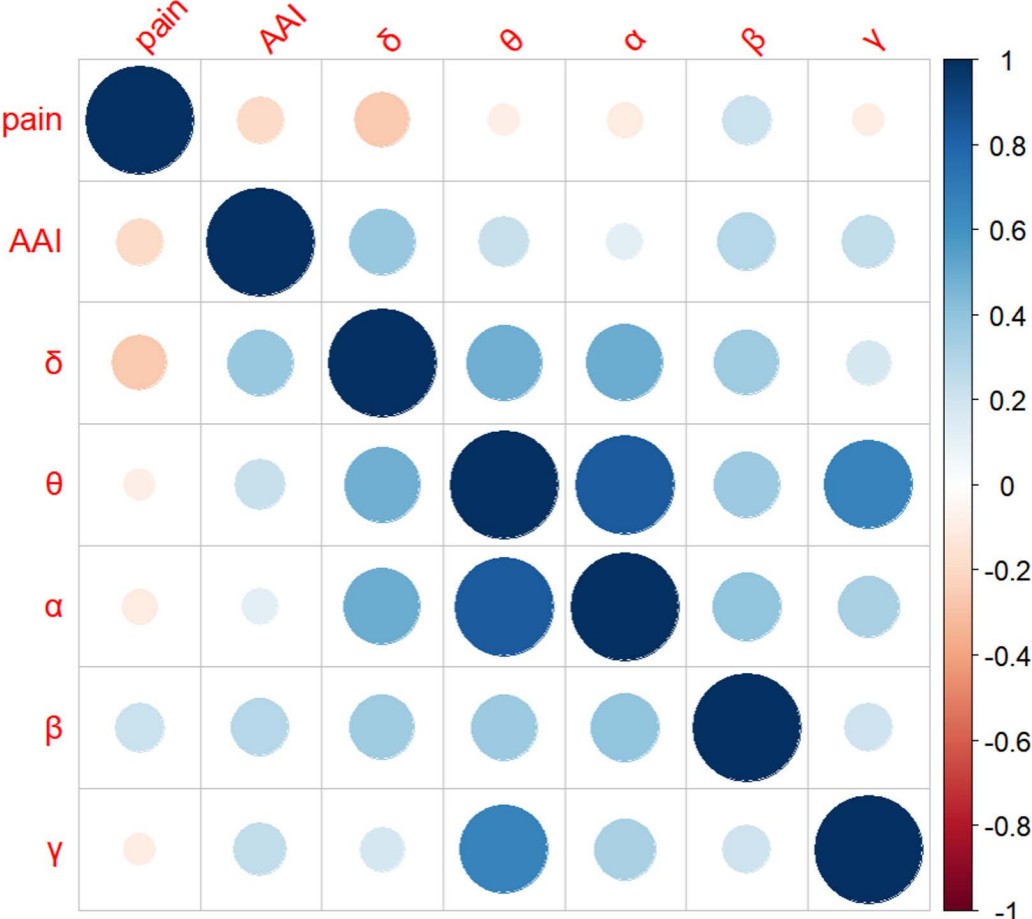

**Fig 2. Correlation matrix of AAI, parietal EEG and pain.**

positively correlated with parietal EEG activity, showing significant associations with both δ frequency (r = 0.28, *p* = 0.02) and β frequency (r = 0.26, *p* = 0.04). No other significant correlations were found outside of those involving EEG frequency bands (Table 3).

### 3.3. The mediating effect of parietal EEG between AAI and pain

Regression analyses were conducted to examine the mediating effects of AAI on pain in HA individuals, with AAI as the independent variable, pain as the dependent variable, and δ, θ, α, β and γ frequencies as the mediating variables. Table 4 shows that the residents' AAI can positively predict the δ frequency (β = 0.28, *p* = 0.02), and simultaneously positively predict the β frequency (β = 0.26, *p* = 0.04) (see Table 5), while no significant relationship was observed with θ (β = 0.23, *p* = 0.07), α (β = 0.11, *p* = 0.38), and γ power (β = 0.18, *p* = 0.16). Among them, when δ frequency was tested as a mediating variable, the total effect of AAI on pain was significant (total effect = −1.20, *p* = 0.04, 95% CI [−2.35, −0.04], excluding 0). The direct effect was not statistically significant (direct effect = −0.85, *p* = 0.15, 95% CI [−2.03, 0.32], including 0), whereas the indirect effect through δ frequency was significant (indirect effect = −0.35, 95% CI [−1.21, −0.01], excluding 0), accounting for 29% of the total effect (see Table 6 and Fig 3). Similarly, when β frequency was examined as the mediator, the total effect remained significant (total effect = −1.20, *p* = 0.04, 95% CI [− 2.35, −0.04], excluding 0). In this case, the

**Table 3. Correlation Analysis of AAI, Parietal EEG and Pain.**

| Factor | M ± SD | 1 | 2 | 3 | 4 | 5 | 6 | 7 |
|---|---|---|---|---|---|---|---|---|
| **1 Pain** | 2.58 ± 1.77 | 1 | | | | | | |
| **2 AAI** | 1.95 ± 0.37 | −0.26* | 1 | | | | | |
| **3 δ** | 2.71 ± 3.55 | −0.31* | 0.28* | 1 | | | | |
| **4 θ** | −1.58 ± 2.60 | −0.07 | 0.23 | 0.57*** | 1 | | | |
| **5 α** | 1.63 ± 4.22 | −0.09 | 0.11 | 0.37** | 0.81*** | 1 | | |
| **6 β** | −7.06 ± 2.30 | 0.17 | 0.26* | 0.31* | 0.51*** | 0.45*** | 1 | |
| **7 γ** | −13.77 ± 3.35 | −0.11 | 0.18 | 0.43*** | 0.61*** | 0.44*** | 0.39*** | 1 |

Note: *$p < 0.05$. **$p < 0.01$. ***$p < 0.001$.

**Table 4. Mediating Effects of EEG Frequencies (δ) Between AAI and Pain.**

| Predictive variable | Outcome variable | β | se | t | p | R | R² | 95%CI | F | p |
|---|---|---|---|---|---|---|---|---|---|---|
| AAI | pain | −1.20 | 0.58 | −2.07* | 0.04 | 0.28 | 0.08 | [-2.35, -0.04] | 2.67 | 0.08 |
| AAI | δ | 2.60 | 1.12 | 2.31* | 0.02 | 0.37 | 0.11 | [0.35, 4.84] | 4.99* | 0.01 |
| AAI | pain | −0.85 | 0.59 | −1.45 | 0.15 | 0.38 | 0.14 | [-2.03, 0.32] | 3.33* | 0.03 |
| δ | | −0.13 | 0.06 | −2.09* | 0.04 | | | [-0.26, -0.01] | | |

Note: *$p < 0.05$. **$p < 0.01$. ***$p < 0.001$.

**Table 5. Mediating Effects of EEG Frequencies (β) Between AAI and Pain.**

| Predictive variable | Outcome variable | β | se | t | p | R | R² | 95%CI | F | p |
|---|---|---|---|---|---|---|---|---|---|---|
| AAI | pain | −1.20 | 0.58 | −2.07* | 0.04 | 0.28 | 0.08 | [-2.35, -0.04] | 2.67 | 0.08 |
| AAI | β | 1.56 | 0.73 | 2.14* | 0.04 | 0.37 | 0.14 | [0.10, 3.02] | 4.87* | 0.01 |
| AAI | pain | −1.51 | 0.59 | −2.58* | 0.01 | 0.37 | 0.14 | [-2.68, -0.34] | 3.22* | 0.03 |
| β | | 0.20 | 0.10 | 2.02* | 0.05 | | | [0.00, 0.39] | | |

Note: *$p < 0.05$. **$p < 0.01$. ***$p < 0.001$.

**Table 6. Effect Decomposition Table Mediated by EEG Delta (δ).**

| | Effect size | se | 95%CI | Relative Effect Size |
|---|---|---|---|---|
| Total Effect | −1.20 | 0.58 | [-2.35; -0.04] | – |
| Direct Effect | −0.85 | 0.59 | [-2.03; 0.32] | 71% |
| Indirect Effect | −0.35 | 0.31 | [-1.21; -0.01] | 29% |

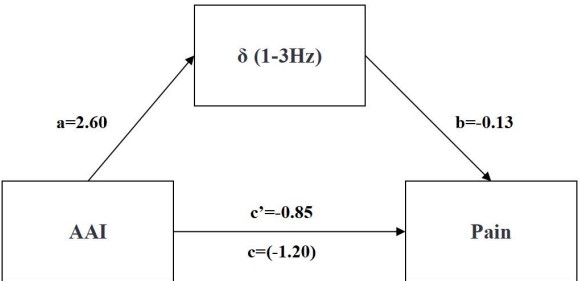

**Fig 3. AAI, δ frequency and pain path map. a*b = Indirect Effect; c' = Direct Effect; c = Total Effect.**

direct effect was also significant (direct effect = −1.51, *p* = 0.01, 95% CI [−2.68, −0.34], excluding 0), and the indirect effect via β frequency was positive and significant (indirect effect = 0.31, 95% CI [0.00, 1.16], excluding 0), representing 21% of the total effect (see Table 7 and Fig 4). These findings indicate that both δ and β parietal EEG frequencies significantly mediated the relationship between AAI and pain.

## 4. Discussion

The findings of this study indicate a significant correlation between AAI, pain, and activity in the parietal cortex. Specifically, parietal δ and β frequencies exhibited a positive correlation with habituation level and a correlation with pain sensation, suggesting that these brain activities serve as a complete mediator in the relationship between AAI and HA pain. According to Zhao et al [67], there is evidence to support the notion that altitude acclimatization/ adaptation may enhance the strength of certain EEG frequency bands within the parietal cortex of individuals. Additionally, individuals residing in HA hypoxic environments demonstrate a heightened reliance on the parietal lobe for the processing of tactile and pain sensations, as noted by Brownsett and Wise [68]. Prior research has indicated that gamma band oscillations induced by short-duration pain tends to localize in central regions, while long-duration and chronic pain are more commonly associated with the prefrontal lobes. This suggests a potential origin of short-duration pain in the primary sensorimotor cortex, and of long-duration and chronic pain in the prefrontal cortex [69], highlighting variations in the brain regions implicated in the modulation of distinct pain types. The correlation between pain experienced at HA and activation of the parietal lobe suggests that the etiology of pain in individuals at HA differs from that of pathological pain or organic somatic lesions.

At HA, respiratory distress resulting from hypoxia is a prominent contributor to pain perception within the organism. In these challenging conditions, pain thresholds are lowered as sensory discrimination heightens [7], indicating heightened pain sensitivity at HA and notable adaptive variations. Those individuals acclimated to the hypoxic conditions of elevated altitudes exhibited reduced somatic pain perception, underscoring the impact of AAI on pain sensitivity.

Significantly, populations residing at HA exhibit varied alterations in low-frequency EEG power bands during acclimatization/ adaptation to the HA environment. Correlation analyses have indicated a negative relationship between certain EEG signals in the parietal cortex and the perception of pain. Specifically, heightened levels of activation in parietal δ

**Table 7. Effect Decomposition Table Mediated by EEG Beta (β).**

|  | Effect Size | se | 95%CI | Relative Effect Size |
|---|---|---|---|---|
| Total Effect | −1.20 | 0.58 | [-2.35; -0.04] | – |
| Direct Effect | −1.51 | 0.59 | [-2.68; -0.34] | 79% |
| Indirect Effect | 0.31 | 0.31 | [0.00; 1.16] | 21% |

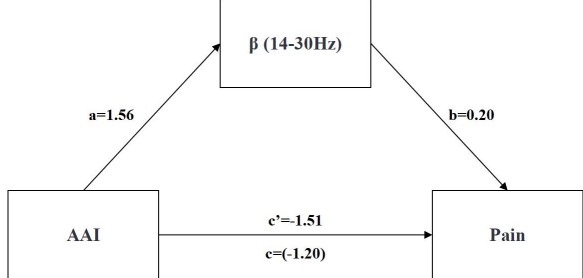

**Fig 4. AAI, β frequency and pain path map. a\*b = Indirect Effect; c' = Direct Effect;c = Total Effect.**

and β power have been linked to decreased pain perception. Unlike performance during acute pain experiences [70], our study did not detect significant changes in α power among individuals at HA. This discrepancy may be attributed to the extended duration of discomfort experienced by participants in our study, lasting over one month, as well as the established relationship between α power activation and conscious relaxation in the typical brain, θ power has been found to be causally related to attention [71], while β power has been associated with vigilance [72–74]. Additionally, the stress induced by HA environments as a stressor has been shown to elicit pain sensations that serve to maintain alertness and prevent relaxation in organisms. The emergence of δ EEG rhythms as a positive indicator of hypoxic brain injury at HA is consistent with previous research [75]. Furthermore, γ power reflects higher mental activity [76], whereas the current study specifically examined the fundamental sensory perception of pain. It is worth noting that γ power predominantly originates from deep brain structures, posing challenges for accurate measurement [77], which may explain the lack of correlation between γ power and pain observed in the present study.

Various types of pain exhibit distinct mechanisms of formation, with the involvement of numerous and intricate brain regions in the perception and regulation of pain. There is no singular neurophysiological or neurochemical alteration that can definitively indicate the presence, absence, or intensity of pain. Furthermore, the subjective evaluation by healthcare professionals and the challenges in effectively communicating pain by individuals contribute to the intricacies of pain assessment, rendering it a challenging endeavor to achieve accurate and unbiased results through a singular approach. Presently, ongoing research is dedicated to further investigating and improving methods for quantifying pain, as highlighted by Li et al [78]. The reliance on subjective self-reports for pain measurement and intervention underscores the inherent limitation in objectively assessing the presence of pain sensations in this study. Future research may benefit from the utilization of detection methods grounded in multi-modal machine learning approaches, which could offer a more objective means of quantifying pain and facilitating differentiated analysis.

## 5. Limitations

In this research, we have made every effort to optimize our experimental procedures. However, as with any scientific investigation, our study presents certain limitations. (1) The cross-sectional design may limit the ability to capture long-term changes or effects. Short-term observations may overlook the dynamic evolution of cerebral activity and pain perception as individuals adapt to HA conditions. Future studies would benefit from adopting a longitudinal approach to better track these changes over time. (2) The sample size was relatively limited, with only 65 participants. This small cohort may constrain the generalizability of the findings. Expanding the sample in future studies would enhance the representativeness of the results and improve the ecological validity of the research. (3) Current understanding of pain perception at high altitude remains incomplete. Specifically, it remains unclear whether the pain reported is localized to specific body regions or more diffuse in nature. To address this, future research should incorporate more precise and high-resolution methodologies to accurately identify and differentiate the bodily sources of pain. Such approaches would allow for a deeper investigation of the underlying mechanisms and distinctive features of HA-related pain, helping to fill existing knowledge gaps and providing a stronger foundation for both theoretical advancement and practical applications.

## 6. Conclusion

A lower level of HA acclimatization/ adaptation indicates a higher intensity of pain perception. As the level of HA acclimatization/ adaptation increases, the activation levels of the delta and beta frequency bands also rise accordingly. Specifically, an upward trend in the activation level of the beta frequency band indicates that residents in HA areas are more sensitive to pain perception. In contrast, an increase in the activation level of the delta frequency band implies a weakened state in the pain perception of HA residents. Additionally, AAI is found to impact pain perception through its influence on the parietal cortex. Therefore, future research on pain in HA environments should take into account alterations in brain function.

## Acknowledgments

The present study owes its existence to the invaluable contributions of numerous participants who, due to considerations of anonymity, are not explicitly acknowledged here by name. Their selfless dedication, exemplified by their willingness to devote time in the laboratory without any form of compensation, is deeply appreciated. Moreover, their openness in sharing highly personal information, which is of significant intimacy to them, is a testament to their commitment to the research endeavor. Without their generosity and participation, the realization of this study would have been unattainable.

## Author contributions

**Conceptualization:** HaiLin Ma.

**Data curation:** Niannian Wang.

**Supervision:** Rui Su, HaiLin Ma.

**Writing – original draft:** Shurong Jia.

**Writing – review & editing:** Hao Li.

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
