## [Decision Letter · Decision Letter 0]

PONE-D-24-39022Parietal-specific activation reveals pain from inadequate levels of altitude acclimatizationPLOS ONE

Dear Dr. Li,

Thank you for submitting your manuscript to PLOS ONE. After careful consideration, we feel that it has merit but does not fully meet PLOS ONE’s publication criteria as it currently stands. Therefore, we invite you to submit a revised version of the manuscript that addresses the points raised during the review process.

We look forward to receiving your revised manuscript.

Kind regards,

Suraj Shrestha, MBBS, DiMM

Academic Editor

PLOS ONE

Journal Requirements:

"This research was supported by Key research and development projects in Tibet Autonomous Region (grant number XZ202201ZY0048G); National Natural Science Foundation Regional innovation development joint fund project of China (grant number U23A20476); the National Natural Science Foundation of China (grant number 32260212); the project of the Tibet Autonomous Region Science and Technology Program “Unveiling the List of Commander-in-Chief”, Key research and development projects in Tibet Autonomous Region (grant number 2023ZYJM001)."

"The author(s) declare that financial support was received for the research, authorship, and/or publication of this article. This research was supported by Key research and development projects in Tibet Autonomous Region (grant number XZ202201ZY0048G); National Natural Science Foundation Regional innovation development joint fund project of China (grant number U23A20476); the National

Natural Science Foundation of China (grant number 32260212); the project of the Tibet Autonomous Region Science and Technology Program “Unveiling the List of Commander-in-Chief”, Key research and development projects in Tibet Autonomous Region (grant number 2023ZYJM001)."

"This research was supported by Key research and development projects in Tibet Autonomous Region (grant number XZ202201ZY0048G); National Natural Science Foundation Regional innovation development joint fund project of China (grant number U23A20476); the National Natural Science Foundation of China (grant number 32260212); the project of the Tibet Autonomous Region Science and Technology Program “Unveiling the List of Commander-in-Chief”, Key research and development projects in Tibet Autonomous Region (grant number 2023ZYJM001)."

5. We note that you have indicated that there are restrictions to data sharing for this study. For studies involving human research participant data or other sensitive data, we encourage authors to share de-identified or anonymized data. However, when data cannot be publicly shared for ethical reasons, we allow authors to make their data sets available upon request. For information on unacceptable data access restrictions, please see http://journals.plos.org/plosone/s/data-availability#loc-unacceptable-data-access-restrictions.

7. Please ensure that you refer to Figure 2 in your text as, if accepted, production will need this reference to link the reader to the figure.

**Additional Editor Comments:**

Dear Authors,

Please find the reviewers comment in the email and attached. Please respond to the reviews clearly and succintly. I hope to receive the revised manuscript.

Thank you.

Reviewer 1

he study explores the relationship between pain perception and acclimatization to high altitudes (HA). Researchers recruited 65 young adults living in Tibet to assess physiological measures (oxygen saturation and hematocrit levels) and brain activity via EEG. Participants were classified into high and low altitude acclimatization index (AAI) groups. Those with higher AAI reported lower pain levels, suggesting better acclimatization reduces pain perception at HA. EEG analysis revealed that parietal lobe brainwaves in delta, theta, and beta frequencies mediate the AAI-pain relationship, with increased frequency power correlating with reduced pain. This highlights the parietal cortex’s role in sensory processing and pain adaptation, offering insights into non-pathological pain management strategies for those living or traveling at HA.

Page 5

2.1 Participants

What altitude is Tibet university ? How long had participants lived at this altitude ? You seem to use the words acclimatization and adaptation as synonyms while they are not. Adaptation refers to a processus taking place over several generations. Did you include participants originating from high altitude areas ?

Did you control for pain medication or other medication that could interfere with pain perception or with altitude acclimatization ?

2.2 Recording equipment

You should specify the number of EEG electrodes (32 I guess).

Why did you chose an eye closed condition over eyes open or both conditions ?

What time of the day did you record EEG ?

2.3 Data pre-processing

You should specify the duration of time periods.

Subsequently, the data was were segmented into time peridos, with the contamined time periods manually labeled and excluded from further analysis analyses.

Page 6

2.5 Pain sensation measurement

You should specify « pain perception or perceived pain »

Page 7

Before the results section, you should had a paragraph describing the statistical analyses that were performed :

main outcome, type of test(s), significance level, power, sample size, etc.

You should had a table detailing :

- age (mean, SD, median, lowest, highest),

- duration of altitude exposure (mean, SD median, lowest, highest)

- AAI score (mean, SD, median, lowest, highest)

- P-FIBS score (mean, SD median, lowest, highest)

- EEG power in each band and total EEG power (mean, SD median, lowest, highest)

3.1 Comparison of differences in pain sensation at the AAI

You performed the analysis on the top 30 and the bottom 30, what happened to the 5 other participants ? According to the table 1, there is no significant correlation between AAI and P-FIBS score. Then, why did you perform regression analyses were conducted to examine the mediating effects of AAI on pain ?

3.3 The mediating effect of parietal EEG between AAI and pain

What do you mean by « marginally significant » ? If your sample size is high enough to reach the suitable statistical power, then if p-value is beyond the cut-off (usually p < 0.05), the result is not significant.

« positive association with δ (β=0.25, p=0.04), θ (β=0.23, p=0.07), and β (β=0.26, p=0.04) » (p=0.07 is not significant)

Page 8

« Consequently, parietal δ, θ, and β frequencies were identified as complete mediators in the association between AAI and pain » : unless I am mistaken, your results do not show a clear association between AAI and pain perception.

Page 10

Discussion

Have you considered including a low altitude control group ?

« This suggest a potential origin of short-duration pain in the primary sensorimotor cortex » : your results show that low HA acclimatization is associated with changes in parietal cortex activity. What reasoning leads you to think that parietal cortex is the source of pain. The changes in parietal cortex activity may as well reflect the processing of pain perception.

I do not understand the last sentence : « Future research may benefit from the utilization of detection methods grounded in multimodal machine learning approaches, which could offer a more objective means of quantifying pain and facilitating differentiated analysis. » What detection methods have you in mind and how do you intend to use them in practice ?

Tables 2 and 3

Do these data come from the whole sample (n=65) or are there any missed data ?

Figure 2

What are the numbers and the numbers between parentheses ?

Reviewer 2

The aim of the present study was to assess the impact of variations in high altitude acclimatization on pain perception in hypoxic environments. A positive association between an altitude adaption index and parietal δ, θ, and β EEG frequencies, as well as a negative correlation between rs-EEG power in these frequency bands and pain perception among high altitude residents is sugested. While the research topic is interesting in its field, the manuscript needs rigorous revision.

Attached you find several comments and concerns that might help the authors improving the quality of their manuscript.

Abstract:

1.) The abstract is still too imprecise in some places. For example, it leaves open the question of why the fast beta waves associated with attention are rated the same as the delta and theta waves, although the delta and theta waves are associated with relaxation, or half-sleep and deep sleep.

2.) It remains unclear in what sense the term "mediators" is used here.

3.) “parietal β (β = 0.26, p = 0.04), δ (β = 0.25, p = 0.04) and θ (β = 0.23, p = 0.07) powers acted as full mediators “: The p-value for Theta power is not significant (p = 0.07). Why do you state it as full mediator in this context? Either the presentation of p-values is confusing here or you should change the meaning of your sentence and take into account that Theta power is not significant.

4.) Please specify "pain" in more detail. What pain locations were recorded? Is it diffuse pain or all types of pain?

Introduction:

1.) Please sum up your hypotheses only at the end of the Introduction section and not in between. In some places the introduction reads more like a collection of ideas than a finished text section. (e.g. page 1: “Therefore, we hypothesize that maladaptive structural alterations in central nervous system (CNS) cells resulting from exposure to plateau hypoxia,”; page 3: “Proposed H1: lower levels of altitude acclimatization are associated with heightened pain sensations.”; And so on)

2.) The introduction is too long and detailed. The introduction would suit better to a narrative review than to an original article. Please shorten the introduction and focus on the most important facts to lead up to your hypotheses.

3.) Why did you only record the EEG for the parietal lobe? Pain processing is usually associated primarily with the cortical areas of the somatosensory cortex, the insular cortex and the ACC. Also the frontal cortex region was found to show changes in pain-related EEG patterns (see Review below). Please explain your decision a little more clearly and name the usual recording locations for pain in the EEG. The following review provides a good overview: https://www-frontiersin-org.translate.goog/journals/neuroscience/articles/10.3389/fnins.2023.1186418/full?_x_tr_sl=en&_x_tr_tl=de&_x_tr_hl=de&_x_tr_pto=rq

4.) In your introduction, you report about many studies assessing for pain perception during acute hypoxia exposure and also for processes during acclimatization periods of several weeks. In your study, however, you are working with a sample of high altitude residents. Please also address this group of people in the introduction. Is there evidence of an increased perception of pain in this population too? Why did you choose this sample and not choose unacclimatized people who were exposed to acute hypoxic conditions and observe them over several weeks?

Methods:

1.) When reading the method section it becomes clear that you not only examined the parietal lobe, but also other cortex areas using EEG. Please make this fact clear earlier (e.g. in the abstract)! Same for Alpha and Gamma waves.

2.) The description of your statistical analyses is completely missing in the Methods section. Please add a corresponding text section!

Results:

1.) Page 8: “The direct impact of acculturation level on pain, in the presence of mediating factors, was determined to be statistically insignificant (p>0.1).”: Did you use a significance level of 0.1 for all statistical calculations? Please specify your approach in a separate text section in the Methods section. Also explain why you did not choose the mostly accepted significance level of 0.05!

2.) Figure 2: Are you showing p-values or r-values here? Or both?

3.) Please explain more detailed the “pain path map” shown in Figure 2.

Discussion:

1.) Please be more structured in your discussion and discuss your findings rigorously. Add a separate text section discussing the limitations of your study!

Reviewer 3

The provided text delves into the intriguing relationship between altitude acclimatization (AAI) and pain perception, particularly focusing on the role of the parietal cortex. The study highlights a significant correlation between AAI, pain intensity, and the activation of specific frequency bands within the parietal cortex.

A key finding is the negative correlation between parietal δ, θ, and β frequencies and pain sensation. This suggests that increased activity in these frequency bands may serve as a protective mechanism against pain. The authors propose that AAI may enhance the brain's ability to process and modulate pain signals, particularly through the parietal cortex.

It's important to note that the study acknowledges the complexity of pain perception and the limitations of relying solely on subjective self-reports. The authors suggest that future research should explore the potential of objective, data-driven methods to quantify pain, such as machine learning approaches.

Overall, this study provides valuable insights into the neurophysiological basis of pain perception at high altitude.

Reviewers' comments:

Reviewer's Responses to Questions

**Comments to the Author**

1. Is the manuscript technically sound, and do the data support the conclusions?

Reviewer #1: Partly

Reviewer #2: Yes

Reviewer #3: Yes

2. Has the statistical analysis been performed appropriately and rigorously? 

Reviewer #1: No

Reviewer #2: No

Reviewer #3: Yes

3. Have the authors made all data underlying the findings in their manuscript fully available?

Reviewer #1: No

Reviewer #2: No

Reviewer #3: Yes

4. Is the manuscript presented in an intelligible fashion and written in standard English?

Reviewer #1: Yes

Reviewer #2: Yes

Reviewer #3: Yes

5. Review Comments to the Author

Reviewer #1: The study explores the relationship between pain perception and acclimatization to high altitudes (HA). Researchers recruited 65 young adults living in Tibet to assess physiological measures (oxygen saturation and hematocrit levels) and brain activity via EEG. Participants were classified into high and low altitude acclimatization index (AAI) groups. Those with higher AAI reported lower pain levels, suggesting better acclimatization reduces pain perception at HA. EEG analysis revealed that parietal lobe brainwaves in delta, theta, and beta frequencies mediate the AAI-pain relationship, with increased frequency power correlating with reduced pain. This highlights the parietal cortex’s role in sensory processing and pain adaptation, offering insights into non-pathological pain management strategies for those living or traveling at HA.

Page 5

2.1 Participants

What altitude is Tibet university ? How long had participants lived at this altitude ? You seem to use the words acclimatization and adaptation as synonyms while they are not. Adaptation refers to a processus taking place over several generations. Did you include participants originating from high altitude areas ? 

Did you control for pain medication or other medication that could interfere with pain perception or with altitude acclimatization ?

2.2 Recording equipment

You should specify the number of EEG electrodes (32 I guess).

Why did you chose an eye closed condition over eyes open or both conditions ?

What time of the day did you record EEG ?

2.3 Data pre-processing

You should specify the duration of time periods.

Subsequently, the data was were segmented into time peridos, with the contamined time periods manually labeled and excluded from further analysis analyses.

Page 6

2.5 Pain sensation measurement 

You should specify « pain perception or perceived pain »

Page 7

Before the results section, you should had a paragraph describing the statistical analyses that were performed :

main outcome, type of test(s), significance level, power, sample size, etc.

You should had a table detailing :

- age (mean, SD, median, lowest, highest),

- duration of altitude exposure (mean, SD median, lowest, highest)

- AAI score (mean, SD, median, lowest, highest)

- P-FIBS score (mean, SD median, lowest, highest)

- EEG power in each band and total EEG power (mean, SD median, lowest, highest)

3.1 Comparison of differences in pain sensation at the AAI

You performed the analysis on the top 30 and the bottom 30, what happened to the 5 other participants ? According to the table 1, there is no significant correlation between AAI and P-FIBS score. Then, why did you perform regression analyses were conducted to examine the mediating effects of AAI on pain ?

3.3 The mediating effect of parietal EEG between AAI and pain

What do you mean by « marginally significant » ? If your sample size is high enough to reach the suitable statistical power, then if p-value is beyond the cut-off (usually p < 0.05), the result is not significant.

« positive association with δ (β=0.25, p=0.04), θ (β=0.23, p=0.07), and β (β=0.26, p=0.04) » (p=0.07 is not significant)

Page 8

« Consequently, parietal δ, θ, and β frequencies were identified as complete mediators in the association between AAI and pain » : unless I am mistaken, your results do not show a clear association between AAI and pain perception.

Page 10

Discussion

Have you considered including a low altitude control group ?

« This suggest a potential origin of short-duration pain in the primary sensorimotor cortex » : your results show that low HA acclimatization is associated with changes in parietal cortex activity. What reasoning leads you to think that parietal cortex is the source of pain. The changes in parietal cortex activity may as well reflect the processing of pain perception.

I do not understand the last sentence : « Future research may benefit from the utilization of detection methods grounded in multimodal machine learning approaches, which could offer a more objective means of quantifying pain and facilitating differentiated analysis. » What detection methods have you in mind and how do you intend to use them in practice ?

Tables 2 and 3

Do these data come from the whole sample (n=65) or are there any missed data ?

Figure 2

What are the numbers and the numbers between parentheses ?

=> See the attached file for more detailed comments

Reviewer #2: The aim of the present study was to assess the impact of variations in high altitude acclimatization on pain perception in hypoxic environments. A positive association between an altitude adaption index and parietal δ, θ, and β EEG frequencies, as well as a negative correlation between rs-EEG power in these frequency bands and pain perception among high altitude residents is sugested. While the research topic is interesting in its field, the manuscript needs rigorous revision.

Attached you find several comments and concerns that might help the authors improving the quality of their manuscript.

Abstract:

1.) The abstract is still too imprecise in some places. For example, it leaves open the question of why the fast beta waves associated with attention are rated the same as the delta and theta waves, although the delta and theta waves are associated with relaxation, or half-sleep and deep sleep.

2.) It remains unclear in what sense the term "mediators" is used here.

3.) “parietal β (β = 0.26, p = 0.04), δ (β = 0.25, p = 0.04) and θ (β = 0.23, p = 0.07) powers acted as full mediators “: The p-value for Theta power is not significant (p = 0.07). Why do you state it as full mediator in this context? Either the presentation of p-values is confusing here or you should change the meaning of your sentence and take into account that Theta power is not significant.

4.) Please specify "pain" in more detail. What pain locations were recorded? Is it diffuse pain or all types of pain?

Introduction:

1.) Please sum up your hypotheses only at the end of the Introduction section and not in between. In some places the introduction reads more like a collection of ideas than a finished text section. (e.g. page 1: “Therefore, we hypothesize that maladaptive structural alterations in central nervous system (CNS) cells resulting from exposure to plateau hypoxia,”; page 3: “Proposed H1: lower levels of altitude acclimatization are associated with heightened pain sensations.”; And so on)

2.) The introduction is too long and detailed. The introduction would suit better to a narrative review than to an original article. Please shorten the introduction and focus on the most important facts to lead up to your hypotheses.

3.) Why did you only record the EEG for the parietal lobe? Pain processing is usually associated primarily with the cortical areas of the somatosensory cortex, the insular cortex and the ACC. Also the frontal cortex region was found to show changes in pain-related EEG patterns (see Review below). Please explain your decision a little more clearly and name the usual recording locations for pain in the EEG. The following review provides a good overview: https://www-frontiersin-org.translate.goog/journals/neuroscience/articles/10.3389/fnins.2023.1186418/full?_x_tr_sl=en&_x_tr_tl=de&_x_tr_hl=de&_x_tr_pto=rq

4.) In your introduction, you report about many studies assessing for pain perception during acute hypoxia exposure and also for processes during acclimatization periods of several weeks. In your study, however, you are working with a sample of high altitude residents. Please also address this group of people in the introduction. Is there evidence of an increased perception of pain in this population too? Why did you choose this sample and not choose unacclimatized people who were exposed to acute hypoxic conditions and observe them over several weeks?

Methods:

1.) When reading the method section it becomes clear that you not only examined the parietal lobe, but also other cortex areas using EEG. Please make this fact clear earlier (e.g. in the abstract)! Same for Alpha and Gamma waves.

2.) The description of your statistical analyses is completely missing in the Methods section. Please add a corresponding text section!

Results:

1.) Page 8: “The direct impact of acculturation level on pain, in the presence of mediating factors, was determined to be statistically insignificant (p>0.1).”: Did you use a significance level of 0.1 for all statistical calculations? Please specify your approach in a separate text section in the Methods section. Also explain why you did not choose the mostly accepted significance level of 0.05!

2.) Figure 2: Are you showing p-values or r-values here? Or both?

3.) Please explain more detailed the “pain path map” shown in Figure 2.

Discussion:

1.) Please be more structured in your discussion and discuss your findings rigorously. Add a separate text section discussing the limitations of your study!

Original ethics review form:

Unfortunately I can not read Chinese letters. Therefore, I cannot comment on the attached document.

Reviewer #3: The provided text delves into the intriguing relationship between altitude acclimatization (AAI) and pain perception, particularly focusing on the role of the parietal cortex. The study highlights a significant correlation between AAI, pain intensity, and the activation of specific frequency bands within the parietal cortex.

A key finding is the negative correlation between parietal δ, θ, and β frequencies and pain sensation. This suggests that increased activity in these frequency bands may serve as a protective mechanism against pain. The authors propose that AAI may enhance the brain's ability to process and modulate pain signals, particularly through the parietal cortex.

It's important to note that the study acknowledges the complexity of pain perception and the limitations of relying solely on subjective self-reports. The authors suggest that future research should explore the potential of objective, data-driven methods to quantify pain, such as machine learning approaches.

Overall, this study provides valuable insights into the neurophysiological basis of pain perception at high altitude.

6. PLOS authors have the option to publish the peer review history of their article (what does this mean? ). If published, this will include your full peer review and any attached files.

**Do you want your identity to be public for this peer review?** For information about this choice, including consent withdrawal, please see our Privacy Policy .

Reviewer #1: No

Reviewer #2: **Yes: ** Mirjam Limmer

Reviewer #3: **Yes: ** Pablo R Morocho Jaramillo

---

## [Author Response · Author response to Decision Letter 1]

1 Mar 2025

Thanks for your reminding, I have replied to your professional suggestions one by one in the document "Response to Reviewers".The part of the fund is for the national scientific research project applied by my professor, since it is not necessary to provide, I have changed the part with the specific contribution of the author.

---

## [Decision Letter · Decision Letter 1]

PONE-D-24-39022R1Parietal-specific activation reveals pain from inadequate levels of altitude acclimatization / adaptationPLOS ONE

Dear Dr. Li,

Thank you for submitting your manuscript to PLOS ONE. After careful consideration and second revision by the reviewers, there are still some concerns need to be addressed by the authors before the final descision is taken. Therefore, we invite you to submit a revised version of the manuscript that addresses the points raised during the review process.

We look forward to receiving your revised manuscript.

Kind regards,

Shimaa Mohammad Yousof, Msc, M.D., Ph.D

Academic Editor

PLOS ONE

Reviewers' comments:

Reviewer's Responses to Questions

**Comments to the Author**

1. If the authors have adequately addressed your comments raised in a previous round of review and you feel that this manuscript is now acceptable for publication, you may indicate that here to bypass the “Comments to the Author” section, enter your conflict of interest statement in the “Confidential to Editor” section, and submit your "Accept" recommendation.

Reviewer #1: (No Response)

Reviewer #3: All comments have been addressed

2. Is the manuscript technically sound, and do the data support the conclusions?

Reviewer #1: Yes

Reviewer #3: Yes

3. Has the statistical analysis been performed appropriately and rigorously? 

Reviewer #1: Yes

Reviewer #3: Yes

4. Have the authors made all data underlying the findings in their manuscript fully available?

Reviewer #1: No

Reviewer #3: Yes

5. Is the manuscript presented in an intelligible fashion and written in standard English?

Reviewer #1: Yes

Reviewer #3: Yes

6. Review Comments to the Author

Reviewer #1: General comment

Thank you for the thorough revision of your manuscript. I appreciate the effort you put into addressing my comments/questions as well as those from the other reviewers. Your detailed and comprehensive responses to all of my questions are commendable, and the changes you made have significantly improved the quality and clarity of the manuscript.

I have only a few remaining minor remarks, mostly related to wording suggestions or slight reformulations that could further enhance the readability of the text.

Please see the attached file for detailed comments.

Reviewer #3: (No Response)

7. PLOS authors have the option to publish the peer review history of their article (what does this mean? ). If published, this will include your full peer review and any attached files.

**Do you want your identity to be public for this peer review?** For information about this choice, including consent withdrawal, please see our Privacy Policy .

Reviewer #1: No

Reviewer #3: No

---

## [Author Response · Author response to Decision Letter 2]

12 May 2025

Dear reviewers and editors! I have submitted the revised manuscript based on your suggestions and made corresponding revisions. I am very grateful for your guidance! Regarding the data in the research results, in addition to the summary statistics, mean, standard deviation, median, etc. are also provided. If any researchers need to obtain the original data, they can contact the corresponding author of this paper.

---

## [Editor Report · Decision Letter 2]

Parietal-specific activation reveals pain from inadequate levels of altitude acclimatization / adaptation

PONE-D-24-39022R2

Dear Dr. Li,

After careful consideration of your manuscript, we feel that it merits publications. Therefore, we are delighted to inform you that your manuscript is accepted for publication in PLOS one journal.

Kind regards,

Shimaa Mohammad Yousof, Msc, M.D., Ph.D

Academic Editor

PLOS ONE

---

## [Editor Report · Acceptance letter]

PONE-D-24-39022R2

PLOS ONE

Dear Dr. Li,

I'm pleased to inform you that your manuscript has been deemed suitable for publication in PLOS ONE. Congratulations! Your manuscript is now being handed over to our production team.

Kind regards,

on behalf of

Associate Professor Shimaa Mohammad Yousof

Academic Editor

PLOS ONE